# Biopolitics and the COVID-19 Pandemic: A Foucauldian Interpretation of the Danish Government's Response to the Pandemic

Philip Højme

The Graduate School for Social Research, Institute for Philosophy and Sociology, Polish Academy of Sciences, 00-330 Warsaw, Poland; philip.hoejme@gssr.edu.pl

**Abstract:** With the coronavirus pandemic and the Omicron variant once again forcing countries into lockdown (as of late 2021), this essay seeks to outline a Foucauldian critique of various legal measures taken by the Danish government to cope with COVID-19 during the first year and a half of the pandemic. The essay takes a critical look at the extra-legal measures employed by the Danish government, as the Danish politicians attempted to halt the spread of the, now almost forgotten, Cluster 5 COVID-19 variant. This situation will serve as a critical point from where to start using Foucault's writings on life and biopolitics in order to expose various legally problematic governmental decisions that became visible during the handling of COVID-19 in general and the Cluster 5 mutation in particular. Reframing the pandemic within Foucault's concept of biopolitics, this essay concludes that the state of exception has led to an increase in biopolitical logic, where some lives have come to matter more than others. As a critical counterpoint to this logic, the conclusion suggests that the notion of biocommunism could provide a suitable reconfiguration of communism. A reconfiguration that could mitigate some of the issues related to biopolitics is uncovered earlier in the essay.

**Keywords:** G. Agamben; J. Butler; biopolitics; biocommunism; COVAX; COVID-19; Denmark; M. Foucault; pandemic; vaccine nationalism





## 1. Introduction

The ongoing global pandemic caused by coronavirus (SARS-CoV-2, hereafter referred to as COVID-19) is perhaps the single most-researched contemporary event of the 21st Century. In the medical field, we have seen an unprecedented acceleration in the development of not only one but a wide array of vaccines to counteract the virus. In addition, it is also worth mentioning the research into various treatments now available for those severely affected by COVID-19 and into the effect that the pandemic has had on society, the economy, and nature. I would tentatively contend that there is not a single academic discipline left untouched by this pandemic. Hence, as a philosopher, I have been pondering how to engage critically with it. The result of this engagement is the following critique of the various laws, ordinances, and restrictions that have been passed to contain (the virus, the infected), decrease (the rate of infection, mortality), or eradicate the virus (something which no longer seems to be a viable option). Hence, this essay seeks to problematize certain aspects of the handling of the COVID-19 pandemic precisely because they have seemingly introduced and multiplied the various techniques often associated with Foucault's notion of governmentality (biopolitics). The increase in regulations aimed at coercing (or nudging) citizens into following health guidelines is, however, problematic, since it has led to groups of citizens reacting strongly against infringements of their ability to choose for themselves. In addition, this essay also problematizes the, often hasty, reaction of politicians which has led to illegal actions being taken by state institutions and politicians, as will be shown with the case of the mink cull in Denmark.

The current proliferation of COVID-19 variants [1] (e.g., D614G, Cluster 5, VOC 202012/01, 501Y.V2), as well as the novel Omicron variant [2], suggests that the virus is here to stay, which in turn means that the measures taken to combat it might also be permanent. There is also the risk of other coronaviruses, such as MERS-CoV (which is 10 times more deadly than COVID-19), making the leap from animals to humans [3]. Because of this risk, there is a need for critical engagement with the actions of governments. To outline this critique, I rely heavily on Foucault's writings, specifically those concerned with health, population, and power [4–7]. Simply put, the aim is to outline a critique of the emergency powers that many governments all over the world have granted themselves in their various attempts to stop the spread of COVID-19. A detailed analysis of even a few countries is beyond the scope of this article; instead, I offer the reader a critique that focuses on the response of the Danish government. However, this essay hopefully outlines a possible critique that others will take as a push toward turning a similarly critical gaze on other national or local responses to the COVID-19 pandemic.

One of the hallmarks of the COVID-19 pandemic has been that medical and industrial hygiene—previously found in hospitals, medical clinics, food processing plants and the medical industry—has invaded spaces previously exempt from conforming to such high standards of hygiene. The new spaces being subjected to these hygiene regimes include public transportation, public squares and offices, and corporate buildings and private homes. In the name of a new-found need for public and private hygiene, governments are using a combination of tightening and relaxing various rules, restrictions, or laws in an attempt to stop the spread of COVID-19.

The Danish decision to instigate a countrywide lockdown was taken relatively quickly and relatively unanimously by the government and parliament. The swiftness with which this decision was made has been suggested as one of the main reasons for the low mortality figures during the 'first wave' of COVID-19 [8–10]. With the country slowly emerging from the grip of the 'second wave' of COVID-19 (in late April 2020), the Danish government once again debated how to proceed with reopening society, especially in light of the realization that, due to the spread of new mutations, a third wave could be on the horizon. However, it seems that even with the vaccination programs rolling out across the affluent world (albeit more slowly in Europe), the spread of various new mutations of COVID-19 seems if not to threaten then at least to be prolonging the current lockdown until sufficient herd immunity has been achieved. From late July 2021, Denmark again saw a high infection rate, partly due to the Delta variant, while December 2021 saw previous records of infection being broken on a daily basis because of the Omicron variant.

In the Danish context, it seems almost symptomatic that most restrictions concerning the pandemic have been or are being framed as 'nudges', rather than as legislation. Regarding the behavior of individual citizens, the preferred strategy appears to have been nudging them towards what has come to be known as *samfundssind* (community spirit). Community spirit can be simplified as 'high levels of trust that the community will do what is necessary for everyone given the circumstances', and as researchers from the Danish HOPE project write in a recently published article, 'interpersonal trust stands out as the most important predictor of response support ... our findings suggest that the highly invasive policies designed to stop the spread of COVID-19 in 2020 tapped into motivations related to collective action' [11] (p. 1152). The following paragraph provides examples of how nudging citizens towards good or healthy behavior played out during 2020–2021 in Denmark. Nudging, in this context, constitutes a form of governmentality insofar as it was a modus operandi through which the Danish government sought to guide or influence the behavior of its citizens.

Firstly, it was not until 12 months after the first lockdown had been imposed (Denmark entered lockdown as one of the first European countries in late February 2020) that the Danish parliament and government finally decided to make it mandatory to isolate for 10 days upon entry into Denmark. At the same time, breaking COVID-19 isolation was made a finable offence for the first time [12]. Thus, for the first 12 months the Danish

government relied heavily on the active cooperation of its citizens with the behavior suggested by government health institutions. Secondly, the decision to cull all mink in Denmark was framed as being based on mandatory rules and legally binding laws, without actually having a legal basis [13] (p. 3). Hence, the fact that Danish citizens were nudged towards acting appropriately stands in stark contrast to how the situation with the Cluster 5 mutation was handled (the Cluster 5 mutation was a COVID-19 mutation primarily found amongst the mink population on Danish farms). The previously 'relaxed attitude' towards the general population thus stands in stark contrast to the il- or extra-legal attitude taken by the government towards the mink farmers and the Danish mink population (a decision that aimed to effectively halt the spread of the Cluster 5 mutation through the effective cull of all mink). That the decision was administered illegally by the Danish police force [14,15] has since led to a parliamentary inquiry, with a report to be published around June 2022.

It is also of interest that a recently published report on the decision to institute a comprehensive lockdown in March 2020 clearly shows the decision to be a political decision that went far beyond the recommendations of the health authorities at the time [16] (pp. 203–224). Thus, one is left wondering why the enforcement of laws and regulations seems to be handled with so little care for the legality of rule of law during the first one to one-and-a-half years of the pandemic. *Could a simple answer to this question be that many of the institutional and legal responses to the pandemic were located in an extra-legal sphere, outside of the usual space governed by parliament and government?*

This answer is, however, not as unprecedented as one might think in Danish realpolitik. One need only mention the collection of laws in Denmark—colloquially known as *Lømmelpakken* [17]—passed in preparation for the protests expected during the COP-15 conference in Copenhagen in 2009. In this context, it is important that, even though some of these laws have subsequently been declared illegal [18], they have not yet been repealed. Therefore, exceptions to the rule of law may not as rare as they are often imagined. Rather, exception is something common in times or situations associated with extraordinary circumstances, such as pandemics and large-scale high-profile international meetings. Extraordinary circumstances like these seemingly invite political leaders to take on powers previously inconceivable within the ordinary understanding of democracy.

With regards to the above comments on the Danish government's response to the pandemic in general and the Cluster 5 mutation in particular, Agamben's book on biopolitics and the pandemic—*Where Are We Now? The Pandemic as Politics* [19]—proposes an interpretation of the Italian response as well as global responses that articulate concerns that are similar to my line of argument. The two most essential aspects, for the purpose of this essay, are the idea that 'bare life separates rather than unites people' [19] (pp. 17–19; 26–30) and the notion of 'biosecurity' [19] (pp. 59–71), a term which links national or state security directly with the health of the population under its control. Hence, the pandemic has not only reduced people's lives from actual lived lives—bios—to biological lives—zoē or bare life [20]. 'The pandemic has [also] shown beyond doubt that citizens are being reduced to their bare biological existences' [19] (pp. 59–71). Because Agamben, both implicitly and explicitly, draws on Foucault's various texts on biopolitics and health, it seems prudent to examine the pandemic in relation to these texts. Thus, the above excursion into Agamben's recent comments on biopolitics, biosecurity, and the pandemic serves as a springboard for the following Foucauldian interpretation of the pandemic. The reduction of life from bios to zoē [20], caused by the proliferation in the use of biopolitical strategies and tactics, calls into question the Foucauldian interpretation itself as it comes up short of suggesting an exit from this situation. Because of this limit, the essay concludes by sketching a reconfiguration of communism, as biocommunism, that draws not only on Agamben's bare life but also on Butler's notion of precariousness [21]. This new notion of communism is skewed away from statism towards individual suffering or individual suffering. This essay thus suggests moving beyond mere identification of the biopolitical logic towards a critique of this logic by providing a sketch for reconciling the needs of individuals with those of 'humanity', or what the biopolitical state calls 'the population'.

As a conclusion to this introduction, I end with a portrayal of the proposed Foucauldian critique through a metaphorical use of a particular geographical location known from von Trier's TV show, *Riget* [22], namely *Blegdamsvej* in Copenhagen, which takes its name from the bleaching industries previously located there between the 17th and 19th centuries. Today, this street houses various facilities: a hospital (*Rigshospitalet, Riget* or *Kingdom Hospital*), the medical faculty of Copenhagen University (*Panum Instituttet* named after Peter Panum, Danish virologist and epidemic researcher), and a prison (*Blegdamsvejens Fængsel*). Metaphorically speaking, *Blegdamsvej* went from being a street where anything sullied was whitewashed to being a street which now houses the technological 'marvels' of the Danish healthcare system (for an account of public healthcare in Denmark, see [23]) and facilities for correcting anti-social behaviors.

With this metaphor, I allude to Foucault's so-called 'biopolitical institutions', which treat humans as statistics—institutions that deal in the 'biopolitics of the population' [4] (p. 139). Hospitals, medical research institutions and prisons all developed from the basis of technologies that sought to guarantee 'the right of the social body to ensure, maintain, or develop its life ... subjecting it [humans] to precise controls and comprehensive regulations' [4] (pp. 136–137). The point of the metaphor is to alert readers to the fact that, for Foucault, hospitals, research institutes, and prisons all serve similar functions. In its own way, each seeks to control, regulate, measure, and maintain, within their social domain, the life of their respective population. And just as von Trier's series Riget provided the viewer with a view of the pre-scientific past upon which modern medicine in general and *Rigshospitalet* in particular were built, so this essay seeks to expose the underlying conditions working beneath the current political concerns with the health of local and global populations alike.

## 2. Emergency Powers and Viral Policing

One particularly interesting suggestion made by Foucault can be found towards the end of the first volume of *The History of Sexuality* [4], where Foucault distinguished between the right of the sovereign (who has the right to kill, or a right to kill) and the aforementioned right of the social body to maintain and develop life (the right to govern biology or a politically governed biology). The change from the former to the latter greatly impacted how the law functions. Whereas the law had previously been punitive, it now became disciplinary. It went from punishing wrongdoers to seeking to correct individuals' divergent behaviors in order to enforce dominant social norms. Therefore, biopolitics and its associated political techniques aim to secure 'the biological existence of a population' [4] (p. 137) and do so by attempting to correct anti-social behavior and physical or mental illnesses.

In addition, the introduction of biopolitics as a technique for governing populations also led to stigmatizing death in order that the state could 'secure the biological existence of their population', whereby death itself became stigmatized, and following this stigmatization, suicide became criminalized and the health of the population—their lives—became a political issue. Thus, death became an affront to the universal power of the biopolitical state because its power was now measured by the health of its population. The normative and statistical standards that allowed a state to determine the health of its population, in turn, led to the development of eugenics and race theories [4] (p. 54). Such a realization must, therefore, lead to the development of a particular critique of the emergency powers granted to the government by the Danish parliament. This application of my critique is vital precisely because of the close association between biopolitics, eugenics, domination, and imperialism that Foucault suggested.

The extra-legal sphere mentioned earlier exists because of the tendency of universal systems to create the possibility of something which does not fit within its borders. This extra or external space becomes thinkable precisely because the legal structure of the state itself facilitates the production of these spaces by declaring individuals who do not comply or conform *persona non grata*. The pandemic seems to have led to a proliferation of these

spaces, simply because of the urgency with which the various countermeasures have been instituted. This much is clear from the non-apology of the chief of the Danish police, who stated that 'the National Police . . . has undertaken a long list of governmental tasks during the coronavirus crisis, often under great time pressure and carrying out tasks which were new and not normally regarded as police work' [24] (my translation). The police chief made this statement because the police had, despite contrary information from the Ministry of Justice, lied to the mink farmers in Denmark about the legality of the nationwide cull. Mistakes such as these have often been covered up by referring to the 'extraordinary' times or the urgency that the pandemic demands, both of decision-makers and those who execute their decisions.

However, this cannot be said of these mistakes. The mistakes are also expressions of what Foucault identified as the unlimited 'objectives of government' [6] (p. 7). Whereas the archaic sovereign was limited by the law of their court and realm, modern biopolitical governments are limited only by political economy and the liberal idea of a "frugal government" [7] (p. 28). The current pandemic, then, once again brings to the fore the question *what is the goal of a 'good' government?* This question is not only asked about the legality of particular official institutions or the government in general, since it is as much asked with regard to the economic utility and profitability of the laws themselves. Therefore, 'if the government happens to . . . go beyond the bounds laid down for it, it will not thereby be illegitimate' [7] (p. 10).

The COVID-19 pandemic has changed societies in ways we cannot yet begin to fathom. During this pandemic, the logic which governs any reasonable government is that a good government balances health and the economy so that the pandemic has as little impact as possible on the economy. The rationale for this is simple: those countries that manage to emerge from the pandemic in a better shape than the rest of the world have the best chance of being the most successful in the post-pandemic world. Moreover, the legal transgressions by the Danish government serve as a reminder that there are no guarantees that governments will abide by a legal framework when they can claim that the emergency has warranted this or that breach of the law. Hence, criminals are no longer those who break laws. Instead, criminals are those whose actions transgress against the sanctity of life. Criminals who commit such offences against our common survival are currently sentenced twice (or more) as severely compared to others [25].

## 3. Healthy Populations: Life, Death, and Vaccine Nationalism

If we interpret the lockdowns, restrictions, and so on as an effect of the stigmatization of death that seems to drive the biopolitical concern for life, then the excess mortality that COVID-19 has inflicted represents an affront to the state. The reason for this is not that the pandemic in itself is an affront to our way of living. Instead, it exposes our government's inability to conquer or overcome nature. Foucault says this in the following lengthy quotation:

> The pressure exerted by the biological on the historical had remained very strong for thousands of years; epidemics and famine were the two great dramatic forms of this relationship that was always dominated by the menace of death . . . [Foucault then states that famine has been conquered in the Western world, but] Outside the Western world, famine exists, on a greater scale than ever; and the biological risks confronting the species are perhaps greater, and certainly more serious, than before the birth of microbiology [4] (pp. 142–143).

Regarding the present, Foucault seems to suggest that the pandemic is, perhaps, something extraordinary and outside the affluent Western world. When, a few months ago, Žižek wrote that 'it's time to accept that the pandemic has changed the way we exist forever' [26], it was not a pessimistic claim per se. Rather than being fatalistic, the claim suggests that there can be no return to normality. The pandemic has forever changed society globally, but the rapid change was, perhaps, only shocking to those parts of the world that had not experienced health crises like these for many years. A Westerner might be shocked by

COVID-19, but those who lived through Ebola, swine flu, or avian flu had never forgotten nature's power. In the sense that I have been referring to this idea, the return of nature should be understood as a trope that aims to highlight how capitalism often seems to have forgotten that humanity and its societies have not immunized themselves to the forces of nature. Humanity is still, in a metaphorical sense, at the mercy of natural forces which, not unlike Odysseus in Homer's *Odyssey*, are subjected to whims beyond the scope of that which 'the administered world' can control.

COVID-19 has unearthed humanity's old bedfellow: death from natural causes. With the 'reappearance' of catastrophic epidemics and pandemics (e.g., Ebola, SARS, swine flu, etc.), humanity's old *nemesis,* nature, once again emerges from the ground upon which we have constructed our modern societies. Thus, humankind in general and Western societies in particular are once again reminded of their insignificant power in the face of death from natural causes. Faced with this predicament, humanity must once again come to terms with its possible end.

The pandemic has also shown us that death is more likely for the poor than for the rich. The ability to prolong one's life, stay home, or be vaccinated is now a privilege more often afforded to the haves than the have-nots. Since the earliest days of the COVID-19 pandemic, this realization has illuminated the stark contrast between the various societal strata and between an affluent West and the rest. Academics, office workers, and public officials have, to a large degree, been able to bunker down at home and keep working, whilst cleaners, clerks, delivery drivers, and medical staff have had to face the storm head-on. Their only protection: face masks and hand-sanitizer.

Foucault noted that 'bio-power was without question an indispensable element in the development of capitalism' [4] (pp. 140–141), and the earlier remark about the increasing dangers of pandemics, famines, and other biological risks also fits well within this claim. The COVID-19 pandemic, a crisis that shows the faults of our societies, has the potential to break the hermetic seal which has protected capitalism from the apparent realization that many of humanity's problems are interconnected with the total structure of this way of living. It is, therefore, not unrealistic that the COVID-19 pandemic may lead to an increase in class consciousness and radical calls for restructuring society. This awareness could perhaps be raised by seeing how the pandemic exposes the underlying logic that has guided many responses to this pandemic. It is a calculating logic where each country seeks to ensure the survival of its population.

The development of more than 50 vaccines [27] is a bleak reminder of this. It reminds us that the value of human life is calculable as simply as X over Y. However, even if possible in theory, neither the COVID-19 pandemic nor the climate crisis has effected solidarity on a truly global scale. And thus even if the pandemic is slowly coming to an end (as of late February 2022, at least in Europe) then the experience has shown that global solidarity was sorely lacking as of 2021. Instead of showing such solidarity, affluent counties were increasingly hoarding available vaccines [28–30] to the detriment of low- and middle-income countries or seeking to protect themselves from the climate changes they have brought about through their previous rapid and extensive industrialization.

This does not mean that governments necessarily prioritize vaccines for investment bankers before the most vulnerable, but it does mean that those living in more affluent countries are likely to be vaccinated before those living in low-income countries. This disparity in access to vaccinations is nothing new [31]. Meanwhile, the former is somewhat tragicomic, as many of these governments at the beginning of the pandemic asked their citizens not to hoard toilet paper, soap, or food. The mantra of the pandemic is the same old saying: money talks and the rich get richer [32].

Pogge, Hoffman, and Hollis [33–35] have suggested a viable way of ensuring both a 'just' distribution and access to medicines, and vaccines via their so-called global *Health Impact Fund*. The goal of this fund would be to offset the global health disparity by instituting various initiatives that seek to offset a particularly medico-economic dilemma: that it is more profitable to cure diseases that affect a smaller population with a high income

than a larger population with a lower income. On the one hand, some commentators, in general agreement with the aim of the *Health Impact Fund*, have argued that 'commitments to human rights . . . place an onus on wealthy countries to ensure that lifesaving vaccines are made available to the poorer countries during crises' [31] (p. 294). The *COVAX* initiative is based on a similar rationale.

On the other hand, some commentators have criticized what could be paraphrased as 'an overly optimistic belief in the altruism of taxpayers in high-income countries' [36], or for 'simplifying the complexity of the issue with measuring the impact of medicines, vaccines, and so on' [37]. While these criticisms raise important concerns related to the *Health Impact Fund* and similar schemes, a thorough examination of this issue is beyond the scope of this essay. The purpose of this digression was simply to highlight a systematic disparity when it comes to access to healthcare. This is important to note, since Foucault's genealogy of biopolitics in *The Birth of Biopolitics* [7], as mentioned earlier, linked the nation state directly to eugenics and racism [5] (p. 50). With the rich world currently hoarding vaccines, it seems increasingly important to draw attention to this realization, especially as it is quite conceivable that governments may start using vaccines as a form of power to gain control over minority populations or other countries.

## 4. Biocommunism and the Logic of Capitalism

Finally, let us return to legal overextension, which is problematic for a state, since it exposes that 'the emperor is indeed naked'. The pandemic has so far shown that those in power were as unprepared to deal with the situation as the rest of us. This problem calls for increased democracy and democratic control in relation to disaster responses. An increase in the democratic control of local responses to the COVID-19 pandemic should delegate actual decision-making power to local communities and allow each community to apply the solution that best fits its situation. One reason why an increase in local decision-making would be beneficial is that the loss of self-determination experienced by individuals during the pandemic could erode public trust in COVID-19 restrictions [38,39]. Moreover, in extreme cases (which are currently multiplying) imposing restrictions upon citizens has led to counter-movements protesting against the various laws and regulations passed by politicians [40].

I therefore propose that 'biocommunism' [41–43] could offer a fruitful framework for reconceptualizing this problem. I wish to move beyond earlier interpretations of biocommunism, however, by suggesting that it be reconsidered as a negative program. As a negative program, biocommunism would, on the one hand, reject both techno-optimism and specific visions of future utopias where humanity's problems are solved through technological breakthroughs [41]. This position is no longer held by the author (unpublished video presentation given in 2020); on the other hand, biocommunism should also reject Wróbel's interpretation as a possible movement 'beyond biopolitics' [42,43].

Biocommunism (as I am able to briefly outline it here) should therefore be reimagined in opposition to both of these earlier conceptualizations. The notion of biocommunism that I suggest is purely a critical position that facilitates a critique of, or critical position towards, the current universal conceptions of life as sacred (the biopolitical paradigm) which have been promoted during this pandemic. Instead, biocommunism would rely on notions such as bare [20] or precarious [21,44] life to guide its critical stance towards the status quo. Thus, biocommunism, according to my preliminary interpretation, draws on Agamben's notion of bare life while simultaneously rejecting his notion that it separates people from one another. Not only this, but biocommunism also stands in opposition to capitalized Communism, e.g., Stalinism, Leninism, and the aforementioned capitalist logic. However, this negative 'program' diverges from Foucauldian philosophy at this instance because the biocommunistic critique of biopolitics suggests more than Foucault's critique of power did. This 'more', which is at the same time, not 'a beyond', is related to biocommunism's reliance on Marx's fourth kind of alienation: species-being, *Gattungswesen* [45]. Nevertheless, because of the refocus on the individual instead of the collective, biocommunism parallels,

at least somewhat, the gist of Foucault's critical philosophy in that they both suggest individuals the kernel from where resistance to biopolitical power (Foucault) or capitalism (Marx) springs forth. Without going into this in detail, this parallel seems to suggest that the abyss between Foucault and Marxism could be bridged by reading Marx's earlier texts as being inclined towards individuals instead of society (i.e., reading Marx closer to anarchism than state communism).

Biocommunism could, precisely because of its rejection of statism as well as capitalism, be a productive conception of a common or even communal humanity that at the same time respects each individual's particularity. This suggestion takes its cue from the dissemination of issues related to the increase in biopolitical solutions to the pandemic. The pandemic has, in one sense, 'united' humanity in a shared struggle. This unity has, nevertheless, been limited so far to national (country-specific) or localized transnational unity (e.g., the European union); where the responses have been transnational they have often been secondary to a prior or primary national response. Hence, the 'unity' has more often than not been a national unity rather than a global unity. On the other hand, this 'unity' has treated individual citizens as populations (in the Foucauldian sense)—as calculable through statistical means—and neglected their individuality by treating each individual as being the same. Hence, biocommunism represents a refocusing on the needs of individuals, in their particular situation, without any recourse to a universal conception of 'life'. As such, it is strongly connected to Marx's fourth kind of alienation, which describes the common life of humanity as related directly to 'the human body' [45] (pp. 75–76). The particularity, the bareness or precariousness of the human body, unifies humanity in its common 'struggle for existence'. By suggesting this, I am not attempting to claim that there is a single universal notion of humanity. I rather mean to suggest that, in humanity's manifold ways of being humans, the one thing that we have in common is our bareness, vulnerability, or suffering. It is thus our shared vulnerability to the pandemic that both unifies and differentiates us—united in that the virus has an impact on our lives, and differentiated in our objective situation, in which access to vaccines, masks, and other protective measures is not the same for all. Our conditio humana could be described as a shared vulnerability to the coronavirus. A biocommunistic critique of capitalist logic, vaccine nationalism, and COVID responses therefore suggests a sensitivity towards individual experiences of vulnerability, of being vulnerable, and of being in a situation where our lives depend on the actions of others.

Leaving this suggestion aside, I now conclude this essay by returning to the realization that capitalist logic underscores most, if not all, of the global responses to the COVID-19 pandemic. I do not attempt to offer a positive program or clear a way out of our predicament. Instead, in line with the general tradition of critical thinking (Critical Theory), the aim of this essay is to provide an occasion for others to continue to rethink and offer critiques of the various responses that the pandemic has prompted.

Examining the state's interest in reproducing the proletariat, Foucault correctly stated that, preceding the biopolitical regime, 'conflicts were necessary (in particular, conflicts over urban space: cohabitation, proximity, contamination, epidemics, such as the cholera outbreak of 1832, or again, prostitution and venereal diseases)' [4] (p. 126). These conflicts were necessary to regulate the population, and hence led to the introduction of the biopolitical regime, which took over the role of nature by sparing 'us' from the overt cruelty of natural randomness (death). We must become aware that we are not necessarily being kept alive out of a humane concern for each of our individual lives. Instead, governments keep their populations alive because they need them as a condition for the state's existence. The biopolitical stigmatization of death (e.g., diseases and natural catastrophes) follows a logic that equates the health and the strength of a state with the vitality of its population. Therefore, we should remember and be aware that beneath the politicians' (often genuine) concern runs the hidden logic of capitalism, which needs an increasing amount of human labor to keep its wheels turning.

At this point in the pandemic, it seems like a good time to have a candid conversation about the interests that led governments to enact their local or specific responses. Particularly since their interest in the health of their populations is a relatively new concern, a concern which enters into a multitude of power constellations in a wide array of institutions, which take on a different form in their specific situation. Thus, a more profound analysis is needed of the police and judicial institutions, healthcare and medical research institutions (both public and private), and governments, capital, and capitalism alike, if we are to come to grips with the brave new post-pandemic world. Such an analysis would aim to show the various constellations of power that government, health, legal, and other institutions entered into and how these in various national, local, or global settings converged around a desire to, or were prompted by the need to, keep the wheels turning—to secure the capitalist mode of production and its associated institutions, as well as an underlying logic and vision of humanity as a homo economicus.

**Funding:** This research received no external funding. The Graduate School for Social Research (GSSR, gssr.edu.pl), of the Institute of Philosophy and Sociology, at the Polish Academy of Sciences has provided organizational support.

**Institutional Review Board Statement:** Not applicable.

**Informed Consent Statement:** Not applicable.

**Data Availability Statement:** Not applicable.

**Conflicts of Interest:** The author declares no conflict of interest.

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
