# Peer review of "Biopolitics and the COVID-19 Pandemic: A Foucauldian Interpretation of the Danish Government’s Response to the Pandemic"

_philosophies, doi:10.3390/philosophies7020034_

Round 1

Reviewer 1 Report

Foucault, Biopolitics and Biocommunism

The article aims to apply Foucault’s notion of biopolitics to criticise the approach taken by Denmark (and other Western countries) in handling the COVID-19 pandemic. 

Central to Foucault’s notion of biopolitics is that the medieval sovereign king had a right to kill, whereas the modern state aimed to discipline the population. Using statistics and other tools developed by the modern state, it was possible to control the population through disciplinary policies. On this view, improving health and living conditions were a means to exercise control over the population by a capitalistic state where the population created a stronger and more prosperous state. 

In the current situation with a global pandemic, the state is continuing to exercise its controlling and disciplinary power over its citizens through the various measures introduced to defeat the virus. Lockdowns, quarantine, and the other measures used by governments are forms of biopolitics for exercising discipline and control over the population to uphold the strength of the capitalistic state.

Denmark’s handling of the pandemic serves as an example in the article of a state using measures without legality and legitimacy and shows how Western countries are using biopolitics during the pandemic to control and discipline the population. A Foucaldian critique of the handling of the pandemic is an original and highly relevant approach to trying to understand the policies employed to defeat the pandemic. Therefore, the article deals with a topic and approach that merits publication.

Nonetheless, by addressing two issues, the article is publishable. Below these are presented. 

1)The central part of the article develops a Foucault-based criticism of how Denmark’s government has handled the pandemic. As an alternative to the centralised disciplinary thinking by governments, the author suggests biocommunism, where it is up to local communities to make democratic decisions about how to fight the spread of the pandemic. It is unclear why communism is the solution to the problem of biopolitics, and it should be developed in more detail why and how these concepts are connected. Secondly, more reasons to substantiate how local democratic decision-making can successfully defeat the pandemic should be needed. The crucial question from a philosophical perspective is why a local democratic approach is normatively justified contra national or global strategies. Local democracy would see a wide range of different measures used to fight the virus, and it is hard to see how this is viable in practical terms, and there are normative arguments for not employing this approach. Because questions like the ones just posed and others are not answered or discussed, the presentation of the solution is underdeveloped and in need of elaboration and more discussion.

2)Another issue raised by the article is the relationship between the pandemic and global climate change. Arguably, both issues are existential threats against humanity, and the author asks why we have not implemented strict measures against climate change since climate change is as existential a threat as the pandemic. The discussion of this issue is on the side of the article’s main point and should be left out. Instead, the article should elaborate on 1) above to argue why communism is a solution to fight the pandemic more persuasively. Climate change and the current pandemic share some similarities, but there are also asymmetries, and it is beyond the scope of the article to compare climate change with the response to the pandemic. Instead, the paper should address in more detail the connection between biopolitics and communism and how communism can overcome the pandemic. 

By addressing these two issues in more detail, the article under review will make an interesting and relevant contribution to understanding the response taken by Denmark and many other countries to fighting the pandemic. 

Author Response

Response to reviewer one’s two comments.

First comment:
1.1 - “It is unclear why communism is the solution to the problem of biopolitics, and it should be developed in more detail why and how these concepts are connected.”

Solution: Added a more detailed account of why biocommunism could solve the biopolitical issues.
Text Added: "

Biocommunism could, precisely because of its rejection of Statism as well as capitalism, be a productive conception of a common or even communal humanity that at the same time respects each individuals’ particularity. This suggestion takes its cue from the dissemination of issues related to the increase in biopolitical solutions to the pandemic. The pandemic has ,in one sense, ‘united’ humanity in a shared struggle. This unity has, nevertheless, been limited so far to national (country-specific) or localized transnational unity (e.g. the European union); where the responses have been transnational they have often been secondary to a prior or primary national response. Hence, the ‘unity’ has more often than not been a national unity rather than a global unity. On the other hand, this ‘unity’ has treated individual citizens as populations (in the Foucauldian sense) – as calculable through statistical means – and neglected their individuality by treating each individual as being the same."

1.2 - “The crucial question from a philosophical perspective is why a local democratic approach is normatively justified contra national or global strategies.”

Solution: Accounted for why local solutions might be better than global.
Answer: "One reason why an increase in local decision-making would be beneficial is that the loss of self-determination experienced by individuals during the pandemic could erode public trust in COVID-19 restrictions (Zahariadis, et al. 2021; Liu, Shahab and Hoque 2022). Moreover, in extreme cases (which are currently multiplying) imposing restrictions upon citizens has led to counter-movements protesting against the various laws and regulations passed by politicians (Gerbaudo 2020)."

1.3 - “Because questions like the ones just posed and others are not answered or discussed, the presentation of the solution is underdeveloped and in need of elaboration and more discussion.”

Solution: See answers 1.1 and 1.2.

2. The reviewer said that the comments about climate change could be left out.

Solution: These comments have been deleted.

The whole text has been proofread by a native English speaker.

Reviewer 2 Report

The article aims to offer an original look at the political and social effects of the current pandemic. The theme is of absolute relevance and interest. The author proposes to use Foucault's work - specifically, his reflections on biopolitics - as a guide to exposing several legally problematic governmental decisions, with particular reference to the action of the Danish government.

The article asks numerous relevant questions, but these are often presented in a fragmented way, without a clear line of argument. Although the introduction highlights the contribution that the article intends to give to a Foucaultian reading of the pandemic and of the actions undertaken by the (Danish) government for its contrast, the text addresses many issues: the issue of the extra-legal nature of government action in the context of a state of exceptionality and emergency declared by itself (following Agamben's reflections); the (international) disparities produced and/or reaffirmed by the measures to combat the pandemic; the consequences that these can have on democratic systems and the need for biocommunism (following the reflections of Žižek); the different 'energy' that governments around the world have put in place to combat climate change and the pandemic;  the theme of biopolitics as a form of governmentality functional to the maintenance and development of the capitalist system. It is my opinion that the article can be greatly improved by illustrating more clearly what the main focus is and by developing the analysis of the relevant topics with greater precision.

Just some examples. Agamben's concept of bare life is sometimes used in a decontextualized way and far from the original elaboration. This is not in itself problematic but requires clarification for the reader. Agamben uses the concept of bare life to refer to the biological fact of living (zoé) by contrasting it with the forms and ways - the meaning - of lived life (bios). In Agamben's reflection, bare life refers to the fact that the pure defence of biological life - bare life - becomes the only goal, to the detriment of the quality and potential of life in its fullness. This action creates 'expendable' lives (homo sacer), i.e., people who anyone (primarily the ruling power) can kill (physically and metaphorically) without committing a crime. Agamben is very critical in observing that the epidemic shows beyond any possible doubt that humanity no longer believes in anything other than the bare existence to be preserved as such at any price. The author instead combines the concept of bare life with that of precarious lives (Butler) to highlight how suffering gives the possibility of new forms of solidarity (see also Žižek and his reading of the pandemic as an opportunity for an option between communism and barbarism).

It is not always clear how the action of the Danish government (more details on the initiatives undertaken would help the reader not familiar with the national situation) is configured as an example of biopolitics and allows to highlight specific aspects of the changes introduced by the pandemic to the democratic system and the capitalist model. The fact that the government acted through 'nudges', rather than legislations is interesting but the article does not fully explain how this is configured as a specific form of governmentality. It is not clear to me what the A. wants to argue by declaring that "Legal measures create the possibility of illegal actions". According to Foucault (and Agamben), it seems to me that the point is precisely that in an emergency and exceptional situation the possibility opens up for those who govern the possibility of defining a new and specific state of legality, which is not based on the previous legality but constitutes itself, in fact, as a new constitutive and institutive legality.

The analysis of the intrinsic contradiction between governmentality inspired by the defence of bare life and governmentality oriented to the defence of homo oeconomicus and, more generally, to the safeguarding of the capitalist system is resolved - perhaps a little too quickly - by superimposing the two governmentalities. In this way, the possibility of grasping the space of tension and the many contradictions that have emerged in the actions of Western governments (but not only) is lost. Perhaps it is precisely from this unresolved tension and conflict that an alternative space for reflection emerges that raises the question of solidarity, the dignity of human life and, in general, the question of a different way of understanding democratic life after the pandemic, having human rights as a compass for intervention.

I would suggest presenting with greater clarity in the Introduction what is the central question to which the article intends to contribute to finding answers, focusing more precisely on the specific object of the contribution. I, therefore, recommend that the contributions that the article makes to a different understanding of governmental dynamics be taken up in the Conclusions in a more precise way.

Author Response

See attached document. The text has also been proofread by a native English speaker.

Round 2

Reviewer 1 Report

The author's response to my comments is convincing and adequate. I therefore recommend publishing the paper in its current form.

Author Response

"The author's response to my comments is convincing and adequate. I, therefore, recommend publishing the paper in its current form."

See above.

Reviewer 2 Report

I'm satisfied with the new version. It is clearer now

Author Response

"I'm satisfied with the new version. It is clearer now"

See above